# Short-Chain Fatty Acids Promote *Mycobacterium avium* subsp. *hominissuis* Growth in Nutrient-Limited Environments and Influence Susceptibility to Antibiotics

**DOI:** 10.3390/pathogens9090700

**Published:** 2020-08-26

**Authors:** Carlos Adriano de Matos e Silva, Rajoana Rojony, Luiz E. Bermudez, Lia Danelishvili

**Affiliations:** 1Department of Biomedical Sciences, Carlson College of Veterinary Medicine, Oregon State University, Corvallis, OR 97331, USA; cadrianocsu@yahoo.com (C.A.d.M.e.S.); rojonyr@oregonstate.edu (R.R.); Luiz.Bermudez@oregonstate.edu (L.E.B.); 2Department of Microbiology, College of Sciences, Oregon State University, Corvallis, OR 97331, USA

**Keywords:** *M. avium*, antimicrobial tolerance, nutrient-limited condition, macrophage infection

## Abstract

*Mycobacterium avium* subsp. *hominissuis* (MAH) is a common intracellular pathogen that infects immunocompromised individuals and patients with pre-existing chronic lung diseases, such as cystic fibrosis, who develop chronic and persistent pulmonary infections. The metabolic remodeling of MAH in response to host environmental stresses or within biofilms formed in bronchial airways plays an important role in development of the persistence phenotype contributing to the pathogen’s tolerance to antibiotic treatment. Recent studies suggest a direct relationship between bacterial metabolic state and antimicrobial susceptibility, and improved antibiotic efficacy has been associated with the enhanced metabolism in bacteria. In the current study, we tested approximately 200 exogenous carbon source-dependent metabolites and identified short-chain fatty acid (SCFA) substrates (propionic, butyric and caproic acids) that MAH can utilize in different physiological states. Selected SCFA enhanced MAH metabolic activity in planktonic and sessile states as well as in the static and established biofilms during nutrient-limited condition. The increased bacterial growth was observed in all conditions except in established biofilms. We also evaluated the influence of SCFA on MAH susceptibility to clinically used antibiotics in established biofilms and during infection of macrophages and found significant reduction in viable bacterial counts in vitro and in cultured macrophages, suggesting improved antibiotic effectiveness against persistent forms of MAH.

## 1. Introduction

*Mycobacterium avium* subsp. *hominissuis* (MAH) is an opportunistic pathogen of the nontuberculous (NTM) group that mainly affects individuals with underlying respiratory disorders (e.g., cystic fibrosis, emphysema) or with immunosuppressed conditions [1,2,3,4]. Healthy individuals are also reported to have MAH infection [1]. The inability of frontline antibiotics to rapidly and effectively eliminate infections has been a chief challenge for treatment of MAH infections [4,5,6], contributing to increased prevalence of NTM globally in the past decade [4]. The therapeutic options for NTM patients are limited [6,7], and the prolonged antibiotic regimens required for MAH treatment can predispose the pathogen for the development of acquired drug-resistance [8]. An equally important challenge in the treatment of MAH infections is the pathogen’s ability to form biofilm in the lung airways of patients [9,10]. The extracellular biofilm matrix creates additional permeability barrier against antibiotics and favors bacterial survival [11]. The biofilm represents an important adaptive mechanism of MAH pathogenesis that facilitates establishment of infections [9,10,12,13,14]. Furthermore, the metabolic remodeling that MAH undergo within biofilms is associated with the persistence phenotype and bacterial tolerance to antibiotic treatment [11,15,16]. The presence of persister cells within biofilms results in treatment relapse as well as it enhances the development of chronic NTM infections [17]. The mechanism of the formation of persister cells in mycobacteria is not well understood. Thus, to increase the therapeutic efficacy of currently used antimicrobials against NTMs and address the treatment challenges of chronic infections, it is crucial to understand the molecular factors promoting mycobacterial transition into persistent phase inside of the host.

Previous studies have demonstrated a link between the low metabolism/respiration and induction of persistence in bacteria of biofilm phenotypes [18,19,20]. The decreased respiration within biofilms has been demonstrated to be result of the limited availability of nutrients and oxygen [18,20,21,22]. The study by Pontes and Groismann revealed that, independent to the physiological mechanism that can promote a non-replicative phase in bacteria, the slow growth rate alone is a crucial factor for the emergence of drug-tolerance in *Salmonella* [23]. The slow grower cells have a lower metabolic rate than actively dividing cells, and this work suggests that the low metabolic activity is essential for enhancing the drug-tolerance phenotype. It has been established for *M. avium* complex (MAC) pathogens that prolonged exposure to deionized water triggers a persistent state associated with a metabolic dormancy and, therefore, bacteria displays significantly lower susceptibility to antibiotics when compared with the actively growing pathogen [24]. Furthermore, study by Greendyke and Byrd revealed that *Mycobacterium abscessus* tolerance to amikacin and clarithromycin is elevated within biofilms and in nutrient-depleted medium [19]. It has been also found that within biofilms *M. abscessus* undergo into a stationary phase characterized with a diminished metabolic rate and tolerance to antibiotics [18,25,26,27]. Moreover, the hostile environment inside the host macrophages, which also imposes a nutrient-limited environment to pathogens [28,29,30], induces the tolerance phenotype in *M. tuberculosis* and *M. marinum* [31] as well as in MAH [15]. Interestingly, study by Lobritz et al. reveals that the capacity of bacteriostatic antibiotics to decelerate cellular respiration prevents bacterial killing by bactericidal antibiotics [32]. In addition, the lack of three major cytochrome oxidase genes in *Escherichia coli* protect bacteria from the lethal effect of ampicillin, gentamicin and norfloxacin. Furthermore, through MAH proteomic study of the biofilm phenotype and in response to antibiotics, our group identified the metabolic shift in oxidative phosphorylation and enrichment in the peptidoglycan synthesis [15].

Recent studies have provided solid evidence that the attenuation of persister cells to antibiotics can be stimulated by activating certain metabolic pathways in bacteria [33,34,35,36,37]. For example, the combination of isoniazid with cysteine can enhance the consumption of oxygen in *M. tuberculosis* and significantly increase the pathogen killing and prevent the formation of persistent cells [37]. In another study, Gutierrez and colleagues discovered that the exhaustion of carbon nutrients coupled with oxidative phosphorylation leads to antibiotic tolerance of stationary-phase cultures of *E. coli*, *Staphylococcus aureus*, and *M. smegmatis* [38].

In the current study, we identified the carbon source-dependent metabolites that MAH is able to utilize within static biofilms and assessed the impact of selected metabolites such as glycerol and short-chain fatty acids (SCFA) on bacterial metabolic activity and growth. We also evaluated the influence of SCFA on MAH susceptibility to clinically used antibiotics within biofilms and infected macrophages.

## 2. Materials and Methods

### 2.1. Bacterial Culture

*Mycobacterium avium* subsp. *hominissuis* 104 (referred as MAH104) isolated from the blood of AIDS patient was used in this study. *Mycobacterium avium* subsp. *hominissuis* 3388 (MAH3388) and *Mycobacterium avium* subsp. *hominissuis* 3393 (MAH3393) clinical strains were isolated from the lung of patients with pulmonary diseases and provided by Barbara Brown-Elliott (University of Texas Health Science Center at Tyler, TX, USA). The MAH104 strain was used in all assays, while MAH3388 and MAH3393 strains only when indicated. Bacteria were cultivated either in 7H10 Middlebrook agar or 7H9 Middlebrook broth medium (Sigma, St. Louis, MO, USA) supplemented with 10% of OADC (oleic acid, albumin, dextrose and catalase; Hardy Diagnostics, Santa Maria, CA, USA) for 7 days at 37 °C. The 0.05% tween 20 was added in 7H9 media where indicated.

### 2.2. Static Biofilm Formation

The static biofilms of MAH104 were generated in the polystyrene 96-well plates as previously described [14] with some modifications. Briefly, the inoculum of 10^9^ colony forming units (CFU)/mL were made in deionized water from bacteria grown on 7H10 agar plates. The suspension was washed three times with deionized water and by centrifugation at 3500 rpm for 20 min. Next, MAH104 pellet was ressuspended in 7H9 medium without 10% OADC, glycerol, tween 20 and tween 80 (non-supplemented 7H9). The suspension was left for 10 min at room temperature to allow clumped bacteria to settle. The top half of the sample was transferred to new tube and adjusted to 1 × 10^8^ CFU/mL using 0.5 McFarland standard and optical density (O.D.) readings. The 100 μL suspension was inoculated in each well of the 96 wells polystyrene plate (BD, Franklin Lakes, NJ, USA) and static biofilms were formed at 37 °C for 7 days or for 14 days when indicated. The crystal violet was added to biofilms at selected time points and washed 4 times with water [39]. The dye was solubilized in 33% of acetic acid and O.D._570ηm_ was determined for control and experimental groups [40]. The O.D._570ηm_ values of experimental wells were subtracted with O.D._570ηm_ of the average of blank wells, which contained only 7H9 broth.

### 2.3. Screening the Capacity of MAH104 Cells to Utilize a Range of Metabolites in Planktonic and Biofilm States

We performed a phenotype metabolic screen for planktonic and biofilm cultures of MAH using Biolog PM1 and PM2A Microplate™ plates that contain 95 various carbon substrates and one negative control (no substrate) on each plate (Biolog™, Hayward, CA, USA). The phenotypic screening was tested simultaneously for planktonic cells and biofilms using the MAH104 inoculum with the same passage number. As per manufacturer’s recommendation, to check the abiotic reaction, we incubated PM1 and PM2A plates for 7 days at 37 °C in IF-0a GN/GP minimal media and appropriate PM additives including Biolog Dye G Mix (100×) but without MAH104. The experiment was made as an end-point assay.

To prepare the inoculum of planktonic cells, MAH104 was cultured in 7H9 medium supplemented with 10% of OADC and 0.05% of tween 20. At the mid-logarithmic phase (O.D._595ηm_ = 0.3 to 0.6), bacterial cultures were harvested and washed three times with deionized water and centrifuged at 3500 rpm for 20 min. MAH104 was incubated in water for 24 h at 25 °C as a starvation step before inoculating in the IF-0a GN/GP media [41]. Fraction of bacterial cells were transferred to a new falcon tube containing 10 mL of IF-0a GN/GP, 120 μL of Biolog Dye G Mix and 1mL of PM additives (24 mM magnesium chloride, 12 mM calcium chloride, 0.0012% zinc sulfate, 0.06% ferric ammonium citrate, 1.2% ammonium chloride, 0.01% tween 20) until it reached to 81% of the percentage transmittance. The 100 μL of this final bacterium suspension was inoculated into PM1 and PM2A plates incubated for 7 days at 37 °C and the O.D._590ηm_ readings were recorded.

For phenotypic screening, MAH104 biofilms were established in IF-0a GN/GP media without PM additive for 7 days. Initially, to solubilize metabolites, we incubated PM1 and PM2A plates with 100 μL of bacteria-free IF-0a GN/GP media supplemented with Biolog Dye G Mix and PM additives for 30 min at 37 °C. Next, the solubilized substrates were transferred into the 96-well plates that contained pre-formed biofilms and incubated for additional 7 days at 37 °C (total of 14 days for the biofilm metabolic assay). To obtain spectrophotometric measurements from biofilms cultures, the plate was centrifuged for 20 min at 3500 rpm and bacteria-free supernatants were transferred into new 96-well plates to reduce bacteria interference with readings at O.D._590ηm_ as per manufacturer’s recommendation. Each test well was compared against negative control well of respective PM plate using unpaired two tailed *t*-test. Signal intensity increase of 25% or more from negative control and a *p* < 0.05 was considered as a positive phenotype.

### 2.4. Testing the Effect of SCFA and Glycerol on Growth of MAH104 Planktonic Cultures in Nutrient-Restricted Media

Planktonic cultures of MAH104 were incubated with propionic acid, butyric acid, caproic acid and glycerol (Sigma) to determine whether utilization of these metabolites can influence bacterial growth in the nutrient-restricted media (non-supplemented 7H9). MAH104 inoculum (1 × 10^8^ CFU/mL) was made from the culture of the mid-log phase grown bacteria on 7H10 agar and 50 μL of suspension was inoculated in 3 mL of 7H9 broth medium alone or supplemented with 0.2% of glycerol, propionic acid, butyric acid and caproic acid in the concentration range of 1–0.01%. Tubes were placed in the shaking incubator under 200 rpm agitation at 37 °C for 12 days, and O.D._595ηm_ readings were taken every 72 h.

### 2.5. Testing the Effect of SCFA and Glycerol on MAH104 Biofilm Cultures

To examine whether consumption of propionic acid, butyric acid, caproic acid and glycerol by MAH104 can affect the biofilm formation process, we performed experiments in non-supplemented 7H9 and evaluated results in sessile MAH104 form of planktonic state (pre-attached planktonic bacteria to polystyrene surface), during static biofilm formation and in established biofilms.

To examine if sessile forms of planktonic MAH104 produce biofilm structures in the presence of selected substrates, 100 µL of bacterial suspension of 1 × 10^6^ CFU/mL prepared in non-supplemented 7H9 medium was seeded in 96-wells polystyrene plates and incubated for 7 days at 37 °C. After, supernatants were removed and surface-attached bacteria were further incubated for additional 7 days in the non-supplemented 7H9 medium with or without 0.05% propionic acid, 0.05% butyric acid, 0.05% caproic acid and 0.2% glycerol. The growth of pre-attached bacteria was monitored every 24 h at O.D._590ηm_, and the biofilm formation was evaluated by crystal violet at day 7 through O.D._570ηm_ readings.

To assess the effect of selected metabolites during MAH104 static biofilm formation, 100 µL bacterial suspension of 1 × 10^8^ CFU/mL prepared in non-supplemented 7H9 medium was inoculated in 96-wells polystyrene plates for 7 days in presence or absence of 0.2% of glycerol and the SCFA at different concentrations (1%, 0.5%, 0.1%, 0.05% and 0.01%). Substrates were added at time zero. The biofilm formation was evaluated by crystal violet at day 7 using O.D._570ηm_ measurements.

Lastly, to analyze the effect of selected metabolites on pre-established biofilms, MAH104 (1 × 10^8^ CFU/mL) was again inoculated in 96-wells plates and biofilms were formed in non-supplemented 7H9 for 7 days. A week later, the supernatant of biofilms was discarded to remove planktonic bacteria and established biofilms were exposed to targeted metabolites (propionic acid, butyric acid, caproic acid and glycerol) for additional 7 days at 37 °C (total of 14 days of biofilm formation). Fourteen-day MAH104 biofilms without targeted metabolites served as a control. Bacterial growth in established biofilms were monitored every 24 h at O.D._590ηm_ over 7 days of exposure to substrates, and the biofilm formation was evaluated by crystal violet at 7 day using O.D._570ηm_. Alternatively, the ability of MAH104 to process tested metabolites (0.05% of SCFA and 0.2% of glycerol) within seven-day biofilms were also examined. The biofilms were established in the IF-0a GN/GP media for 7 days and then 100 μL of IF-0a GN/GP media with selected metabolites or only non-supplemented 7H9 (negative control) were added and incubated at 37 °C for additional 7 days. The day in which the metabolites were included in the established biofilms was considered as day 0 of incubation. The Biolog Dye G Mix was added at day 0, 4 and 7 for 24 h and incubated at 37 °C. The spectrophotometric measurements were obtained at O.D._590ηm_ after centrifuging plate for 20 min at 3500 rpm and transferring bacteria-free supernatants to a new 96-well plate to reduce bacteria interference with readings.

### 2.6. Antibiotic Treatment of MAH Biofilms

To assess MAH104 tolerance to antibiotic treatment within biofilms, we formed bacterial biofilms for 14 days as described above. The supernatants were gently removed and replenished with a fresh non-supplemented 7H9 medium containing either amikacin (4 μg/mL) or clarithromycin (16 μg/mL) and in combination or without combination of targeted metabolites (0.05% propionic acid, 0.05% butyric acid, 0.05% caproic acid or 0.2% glycerol). As a control, biofilms were incubated only with non-supplemented 7H9 medium without any antibiotic addition. MAH104 biofilms that were exposed with only 0.05% propionic acid, 0.05% butyric acid, 0.05% caproic acid or 0.2% glycerol in the non-supplemented 7H9 medium served as an additional control. The established biofilms were disrupted with 100 μL of 0.02% Triton X-100, serially diluted and plated on the 7H10 agar plates for bacterial CFU counts. We also tested the ability of SCFA to enhance the efficacy of amikacin against biofilms formed by the MAH3388 and MAH3393 lung isolates. These strains are resistant to clarithromycin due to the presence of *erm* gene [42] and, thus, they were not subjected for clarithromycin treatment assays.

### 2.7. MAH104 Infection and Antibiotic Treatment of Human Macrophages

Human THP-1 monocytes (TIB-202) (American Type Culture Collection, Manassas, VA, USA) were maintained in 75 cm^2^ tissue culture flasks and in the RPMI-1640 medium supplemented with 10% of heat-inactivated fetal bovine serum (FBS, Gemini Bio-products, Sacramento, CA, USA) at 37 °C with 5% CO_2_. Prior experiments, THP-1 monocytes were differentiated into macrophages in 48-well plates using 20 ng/mL phorbol 12-myristate 13-acetate (PMA; Sigma-Aldrich, St Louis, MO, USA). The antibiotic killing assays were performed as previously described to record the intracellular MAH104 survival rate [15]. Macrophage monolayers were incubated with MAH104 inoculum at Multiplicity of Infection (MOI) 10 bacteria: 1 cell, extracellular bacteria were removed by washing cells 3 times with HBSS and then through 400 μg/mL amikacin treatment for only 1 h. Infected THP-1 macrophages were subsequently treated with either amikacin (4 μg/mL), clarithromycin (16 μg/mL), antibiotic (amikacin or clarithromycin) with targeted metabolites (0.05% propionic acid, 0.05% butyric acid, 0.05% caproic acid or 0.2% glycerol) or only with the targeted metabolites. The infected THP-1 monolayers without antibiotic and metabolite treatments served as a control. Macrophages were replenished with fresh media containing antibiotics or no antibiotics every other day. Cells were lysed at 2 h (baseline) and day 4, and the number of viable bacteria was determined by CFU counting on 7H10 agar plates.

### 2.8. Statistical Analysis

The effect of SCFA and glycerol in biofilm formation was examined by unpaired two tailed *t*-test between tested and control groups. This statistical test was also used to assess the capacity of planktonic and MAH biofilms to metabolize SCFA and glycerol. The growth of MAH in the presence of tested substrates as a planktonic and sessile state as well in biofilms were compared through Two-way ANOVA analysis, followed by bonferroni post-test. The unpaired two tailed *t*-test was applied for comparing the antibiotic treatment efficacy in established biofilms as well as during tissue culture infection in the presence or absence of SCFA or glycerol. The statistical analysis and graphical outputs were made in GraphPad Prism software (version 6.0).

## 3. Results

### 3.1. MAH104 Displays a Decreased Capacity for Processing of Carbon Substrates in Biofilms

In order to determine which carbon substrates MAH104 is capable to consume in biofilms, in the initial testing, we used carbon substrates of the Biolog Phenotype Microarray PM1 and PM2A plates. The experiment was made as an endpoint assay, and results obtained for both biofilms and planktonic cultures are shown in the Table 1, Figure 1 and Appendix A. Our data indicate that while planktonic bacteria are able to process 16 out of 190 substrates, biofilm cultures of MAH104 utilize 11 substrates (Table 1). Within the set of metabolites used by planktonic cells, biofilms were unable to metabolize α-keto glutaric acid, α-keto butyric acid, α-hydroxy butyric acid, acetoacetic acid and monomethyl succinate. Furthermore, MAH104 of the biofilm state showed a significantly lower capacity to utilize glycerol (11.14-fold), tween 20 (5.4-fold), tween 40 (3.8-fold), acetic acid (2.55-fold), tween 80 (3.45-fold), propionic acid (5.49-fold), methyl pyruvate (1.8-fold), pyruvic acid (2.21-fold) and (4.27-fold) when compared to planktonic bacteria (Figure 1). There was no significant difference in utilization of butyric acid (biofilm O.D._590ηm_ = 0.430 ± 0.091, planktonic O.D._590ηm_ = 0.289 ± 0.083, *p* = 0.31) and sebacic acid (biofilm O.D._590ηm_ = 0.411 ± 0.059, planktonic O.D._590ηm_ = 0.210 ± 0.069, *p* = 0.092) by bacteria in both states.

### 3.2. Glycerol and SCFA Support the Growth of Planktonic MAH104 in Nutrient-Limited Media

We selected the SCFA substrates (propionic acid, butyric acid and caproic acid) and glycerol to evaluate the MAH104 metabolic state in the nutrient-limited media when supplemented with selected carbon sources. The SCFA and glycerol were chosen for further investigation based on their potential to influence the outcome of mycobacterial infections [43,44,45,46,47]. The SCFA and glycerol were commonly utilized by both planktonic and biofilm bacteria. While butyric acid was equally utilized by both planktonic and biofilm cultures of MAH104, the propionic acid and caproic acid were highly metabolized by planktonic cells than MAH104 of biofilm culture. In addition, glycerol exhibited a similar profile that was observed for propionic acid and caproic acid.

Since concentrations of selected metabolites presented in the Biolog Microarray plates are unknown, at first, we established the optimal concentrations of propionic acid, butyric acid and caproic acid that induced the MAH104 growth in non-supplemented 7H9 media. Glycerol at 0.2% concentration is routinely used for cultivation of mycobacteria. The experiments were carried out in non-supplemented 7H9 medium without any carbon sources, no OADC, no glycerol and no tween 20 or 80 (Figure 2). As expected, 0.2% glycerol was able to support MAH104 growth in non-supplemented 7H9 when compared to bacterial growth in the non-supplemented 7H9 (Figure 2A). Results revealed that planktonic MAH104 was able to grow in non-supplemented 7H9 when it was supplied with propionic acid, butyric acid and caproic acid at 0.05% concentration (Figure 2B–D).

### 3.3. The Sessile MAH104 of Planktonic State Displays Higher Capacity for Growth and Biofilm Formation when Incubated with Glycerol or SCFA

To tested if selected metabolites could induce the growth of sessile forms of MAH104 in the nutrient-limited media, the low concentration of planktonic bacterial inoculum (10^6^ CFU/mL) were seeded in non-supplemented 7H9 media for 7 days at 37 °C to allow cells for attachment on the polystyrene surface. A week later, supernatants that contained planktonic cells were discarded and surface-attached bacteria were further exposed to nutrient-limited media (non-supplemented 7H9) with or without glycerol and SCFA for 7 days. MAH104 growth was followed by O.D._595ηm_ and biofilm formation was determined using the crystal violet staining (O.D._570ηm_) (Figure 3). We did not observe the growth of sessile MAH104 in non-supplemented 7H9 broth, however, propionic acid, butyric acid, caproic acid and glycerol enhanced bacterial growth (Figure 3A). Interestingly, the biofilm assay performed for the sessile MAH104 in a media supplemented with substrates demonstrated higher readings (propionic acid, O.D._570ηm_ = 0.25 ± 0.01; butyric acid, O.D._570ηm_ = 0.32 ± 0.02; caproic acid, O.D._570ηm_ = 0.177 ± 0.017), and recorded values were significantly higher than O.D._570ηm_ obtained from the control (O.D._570ηm_ = 0.091 ± 0.007) (*p* < 0.01) (Figure 3B). In addition, it is also important to highlight that O.D._570ηm_ values obtained at day 7 for the sessile MAH104 control exposed with the non-supplemented 7H9 medium in absence of substrates (O.D._570ηm_ = 0.11 ± 0.039) showed no significant difference when compared with the same sessile MAH104 control at day 14 (O.D._570ηm_ = 0.091 ± 0.007) (*p* = 0.37), suggesting no biofilm formation.

### 3.4. Glycerol but Not SCFA Promote MAH104 Growth in Biofilms

To evaluate if selected substrates can promote bacterial growth within biofilms, we established static biofilms in presence of 0.2% glycerol and in wide-range concentrations of propionic acid, butyric acid, caproic acid (Figure 4). MAH104 cultures in the non-supplemented 7H9 served as a negative control. The Figure 4A demonstrates a significant increase in the biofilm formation during MAH104 incubation with glycerol (3.2-fold higher) in comparison to the control without a supplement. Interestingly, MAH104 incubation with high doses (1% and 0.5%) of propionic acid, butyric acid and caproic acid did not promote biofilm formation (Figure 4B–D, respectively) and was significantly lower when compared with the corresponding substrate treatments at lower concentrations. In contrast, biofilm formation occurred in presence of 0.1%, 0.05% and 0.01% concentrations of the SCFA but there were no significant differences between the experimental (substrate) and control groups.

In addition, the effect of SCFA and glycerol were tested on pre-established biofilms, and MAH104 growth was also recorded in presence or absence of targeted substrates (Figure 5). As shown in the Figure 5A, MAH104 displayed a significant increase in O.D._595ηm_ values for glycerol when compared with the pre-formed biofilm control over 7 days. Likewise, a significant increase in the biofilm formation was observed in the glycerol tested group versus the control (glycerol O.D._570ηm_ = 1.562 ± 0.1751; control O.D._570ηm_ = 0.332 ± 0.073; *p* = 0.0029) (Figure 5B). Furthermore, pre-established biofilms incubated with glycerol exhibited a significant biofilm formation when compared with the propionic acid, butyric acid and caproic acid tested groups (glycerol O.D._570ηm_ = 1.562 ± 0.1751; propionic acid O.D._570ηm_ = 0.3855 ± 0.048; butyric acid, O.D._570ηm_ = 0.490 ± 0.121; caproic acid O.D._570ηm_ = 0.383 ± 0.065). Altogether, results indicate that glycerol but not SCFA promote MAH growth and increase the biofilm mass. In addition, we examined MAH104 physiological status through measuring the oxidation rates over 7 days and found that while glycerol was only substrate to promote bacterial growth, both glycerol and SCFA notably increased metabolism of MAH104 within established biofilms when compared with the non-metabolite treated 7H9 control group (Figure 5C).

### 3.5. Glycerol and SCFA Enhance Antibiotic Efficacy against MAH104 in Biofilms and in Cultured Macrophages

Our results demonstrate that glycerol and the SCFA are processed by MAH104 in biofilms as an energy source, and these substrates promote bacterial growth in nutrient-restricted media. In addition, glycerol induces MAH104 growth in biofilms. Due to the fact that SCFA and glycerol increase the metabolic and replication rate of mycobacteria in conditions that is also known to promote the drug-tolerance phenotype, we hypothesized that by activating the metabolic/growth state of MAH104 in biofilms, we may increase bacterial susceptibility to antibiotics as well.

To determine if glycerol, propionic acid, butyric acid and caproic acid can improve the efficacy of clarithromycin and amikacin against MAH104, first we performed in vitro assay on established biofilms. While the experimental biofilm groups were cultured with targeted metabolites and treated with antibiotics, the control group was exposed to antibiotics in absence of substrates. We also investigated whether SCFA could improve the amikacin efficacy against biofilms formed by the clarithromycin-resistant strains MAH3388 and MAH3393. These strains were not subjected for clarithromycin treatment assays. Table 2 displays CFUs counts for three MAH clinical strains (MAH104, MAH3388, MAH3393) after clarithromycin and/or amikacin treatment in established biofilms, displaying the significant reduction in viable bacterial number within the substrate added antibiotic experimental groups.

Since it is well known that intracellular environment of the macrophage enhances the development of the drug-tolerance phenotype in mycobacteria [15,31], next, we evaluated MAH104 killing by antibiotics in cultured THP-1 human macrophages that were supplemented with glycerol or SCFA. Data presented on Table 3 demonstrate that while MAH104 grew intracellularly over four days in THP-1 cells, the treatment with both antibiotics delayed bacterial growth for more than 1-log. Differences between clarithromycin and amikacin treated experimental and non-treated control groups of glycerol, propionic acid, butyric acid and caproic acid were found to be significant (*p* < 0.05).

## 4. Discussion

The facultative MAH can gain entry into the host through the respiratory or gastrointestinal tract and, in the absence of adequate immune responses, successfully establish the infection [4]. MAH infections require combinational and prolonged therapy and, due to high tolerance or resistance to antibacterial treatment, the rate of the favorable outcome in patients has been reported in only 40% to 60% [5,48]. Within the host, under extended exposure of antibiotics and environmental stresses, MAH undergo cell surface modifications and metabolic remodeling, which results in the development of persistent phenotype [15]. This tolerance state of the pathogen directly influences the effectiveness of antibiotics in vivo, subsequently, causing chronic MAH infections [49]. The experimental evidence in vitro suggests that, in the hostile environments such as nutrient-limitation and lack of the oxygen, MAH survives through rapid shift from aerobic to anaerobic respiration [50] and induction of a nonreplicating state [24]. In addition, in early colonization of the host, MAH develops microaggregates (pre-biofilm form) that are more proficient at binding to and invading epithelial cells [10,12] and influence the respiratory infection in vivo [9]. During MAH chronic infection in patients with underling lung conditions such as cystic fibrosis and chronic obstructive pulmonary disease (COPD), the pathogen forms robust extracellular biofilms in bronchial airways of the lung [51]. Biofilm structures prevent the optimal penetration of antimicrobials, leading to decreased potency of antibiotic action and, in addition, the subsequent physiological remodeling of MAH creates added challenge, reducing the susceptibility to antibiotic treatment [15].

The ability of mycobacteria to utilize SCFA has been established by other groups [52,53] but the use of these metabolites by MAH in contexts that is relevant for human infection (e.g., biofilms, infected macrophages) has been only few explored [43]. Current report is the first to describe processing of SCFA by MAH104 biofilms. Studies suggest that the nutritional requirement, which affects bacterial growth during limited conditions and also derives physiological transformation processes, can be exploited and used as a new therapeutic strategy to stimulate bacterial metabolism and increase the efficacy of existing antibiotics. In the present research, we tested approximately 200 exogenous carbon source-dependent metabolites to identify substrates that MAH104 was capable of processing when in biofilms. We have identified significant increase in substrate oxidation rates for glycerol and SCFA (propionic acid, butyric acid, and caproic acid) in MAH104 biofilms when compared with the non-substrate control. The oxidation rates for SCFA in biofilms were found to be comparable to glycerol. It is also important to highlight that for the majority of substrates tested the oxidation rates in MAH104 were lower in biofilms than in planktonic state, which is in agreement with the reduced metabolic activity of bacteria within biofilms [20] and with a recent report that showed a significant inhibition of the butanoate and propanoate metabolic pathways in MAH104 biofilms [15]. There is no data available on caproate metabolism in MAH biofilms, and this study is the first report to validate the important role of caproate metabolic pathway in mycobacteria biofilms.

Here, we established that higher concentrations of SCFA impair biofilm formation and the growth of planktonic MAH104, while a lower concentration of these compounds not only increase the metabolic activity of established biofilms but also promote the growth of planktonic MAH104 in nutrient-limited conditions. SCFA, however, did not promote growth of mycobacteria within established biofilms. Moreover, lower amount of the SCFA enhanced bacterial growth and biofilm formation by sessile forms of MAH104 in nutrient-limited conditions. These results reveal that SCFA effect on mycobacteria dependents on the substrate concentration. Also, the physiological state of MAH104 seems to have an influence how mycobacteria utilize SCFA, for example, at later stages of biofilm (established biofilms) MAH104 exploit these substrates for metabolic respiration but not for bacterial growth. In contrast, SCFA stimulate MAH104 growth in early biofilms (sessile MAH). Altogether, we demonstrate that low concentration of SCFA enhance the metabolic activity of MAH104 in conditions that otherwise will trigger bacterial tolerance phenotype to antibiotics [18,21,24,54].

A large body of evidence indicates a direct link between metabolic activity and antibiotic susceptibility, and improved antimicrobial activity has been demonstrated for several class of antibiotics to tie-up with enhanced metabolism [55]. For example, alanine and glucose metabolites significantly increase bactericidal effects of aminoglycoside treatment against *K. pneumoniae*, *S. aureus*, *P. aeruginosa,* and drug resistant and biofilm and persister forms of clinical isolates of *Edwardsiella trada* [55,56]. The exogenous metabolites such as glucose and fructose boost the aminoglycoside killing persisters of *S. aureus* and *Escherichia coli* and within biofilms via activation the central metabolic pathways [34]. In addition, while a recent finding reveals that bacterial slow growth phenotype has direct relation with a drug-tolerance [23], it is generally accepted that actively dividing cells exhibit a higher susceptibility to antibiotics than non-dividing cells [18,57,58,59]. Based on these findings and due to the fact that MAH becomes physiologically active within established biofilms during exposure to propionic, butyric and caproic acids, we evaluated susceptibility to clinically used antibiotics in presence of SCFA. We found that all three fatty acids significantly induced killing of MAH104 by amikacin and clarithromycin in biofilms when compared with the viable bacterial number treated with only antibiotics or exposed only with metabolite supplements. In addition, we show that the SCFA compounds used in our study improved the efficacy of amikacin against the clarithromycin-resistant strains MAH3388 and MAH3393. Furthermore, previous studies from our group showed that the metabolic pathways of butanoate and propanoate are down-regulated in mycobacterial biofilms [15,60] and, in this study, we demonstrate that stimulation of these metabolic pathways increase susceptibility of MAH to antibiotics. Thus, the butanoate and propanoate metabolic pathways are potentially linked with MAH tolerance/susceptibility in biofilms.

Due to the fact that intracellular pathogens develop persistence phenotypes within the host and cellular environmental stresses play significant role in this process [61,62,63], we examined MAH susceptibility to antibiotic treatment in combination with SCFA within macrophages. The treatment with both antibiotics was found to be more lethal in combination with SCFA, significantly inhibiting intracellular MAH104 growth within the host macrophages. To highlight, the CFU numbers of viable intracellular MAH104 were similar for infected macrophages in presence or absence of SCFA, indicating that MAH104 do not use these substrates for intracellular growth. Since our results show that antibiotics significantly increase MAH104 killing in presence of SCFA and due to the fact that several studies have also demonstrated enhanced bacterial growth or cellular respiration by SCFA [23,32,34,35], here we propose that the SCFA increase the respiration of MAH104 inside macrophages and, thus, promote efficient bacterial killing. We need to also highlight that *M. tuberculosis* is capable of utilizing propanoate during infection of macrophages and this process triggers the drug tolerance mechanism [44]. Together, current work demonstrates that metabolic stimuli generated by propionic, butyric and caproic acids boosts mycobacterial metabolism and significantly influence antimicrobial activity of clinically relevant drugs in biofilm and microaerophilic (intracellular) states of MAH104 ([15] and this study). We also need emphasize that within the host environment mycobacteria (whether it is in the extracellular or intracellular state) are expose to more complex milieu such as anaerobic condition, high/low osmolarity and immune defenses generated by other immune cells shaping macrophage activation phases. These cellular processes may influence the study outcome. Few MAH clinical strains tested here are also potential limitations of this study.

Our results show that glycerol promotes MAH104 growth at early and later biofilm stages. In contrast to SCFA, glycerol enhanced the metabolic activity of MAH104 in all conditions tested, and this study is the first attempt to demonstrate that stimulation of glycerol metabolism in MAH104 can improve the efficacy of antibiotics. Prior studies on *M. tuberculosis* research have indicated that the catabolism of glycerol increases anti-tuberculosis drug activity in vitro and in vivo, while defects in glycerol catabolism were associated with drug tolerance [64,65]. Furthermore, a gradual depletion of glycerol or glucose in *M. tuberculosis* culture leads to the phenotypic remodeling of bacteria linked with a persistence [54]. The presence of glycerol in mice lungs suggests that mycobacteria may be exposed to this substrate in vivo [65]. In addition, the mutation in the *prpR* gene, a transcription regulator of propionate metabolism [66], confer a conditional multi-drug tolerance of *M. tuberculosis* clinical strains when bacteria are exposed to propanoate or during infection of macrophages [44]. Thus, while considering the use of glycerol and SCFA for improvement of mycobacterial treatment, we should also reflect the potential of MAH strains bearing mutations in genes of glycerol and SCFA metabolism and possibly may mediate tolerance mechanism.

This study also highlights the necessity of future research to explore a link between changes in gut microbiota and MAH infection. The SCFA constitute the most abundant metabolites in the gut microbiota and possess several roles in microbiome-host signaling (e.g., metabolic, immune) and also play a crucial role in human health and disease [67]. Recent metabolomics studies on identification of potentially important microbial metabolites derived from the gut of diseased individuals established numerous metabolites including SCFA, contributing to disease development [68,69]. SCFA has been shown to affect *M. tuberculosis* but the role of these metabolites for MAH infection is unknown. Previous findings indicate that higher abundance of propionate and butyrate producer’s bacteria are connected with the development of tuberculosis by increasing anti-inflammatory responses in the host [45,46,47]. It has also been shown that metabolic products of anaerobic fermentation like butyrate and other SCFA impair macrophage ability to phagocytize and kill *M. avium* complex [43]. SCFA also carry anti-inflammatory activity, and propionate was shown to increase susceptibility to *Klebsiella pneumoniae*, *Staphylococcus aureus* and *Candida albicans* infections by dampening innate immune responses in vitro via inhibition of cytokine and NO production; however, it did not display a major impact on passive or natural immunization and on susceptibility to infections in vivo [70].

The existing evidence supports the notion that SCFA contribute to cystic fibrosis-specific alterations toward inflammation that benefit pulmonary colonization of *Pseudomonas aeruginosa* in bronchial airways [71]. In contrast, some studies indicate the protective role of SCFA against infections [72,73,74]. A recent discovery by Hu Y *at al*. underlines a fact that the significant reduction in bacterial species of gut microbiota that are responsible for SCFA synthesis, including propionate producers, can favor the development of tuberculosis disease [72]. Furthermore, the synergistic ability of caproic acid has been suggested to suppress the expression of invasion genes of *Salmonella* serovar Enteritidis and influence bacterial colonization and growth in vivo [74]. Interestingly, the cefoperazone treatment significantly reduces butyric acid, propionic acid and acetic acid levels in the cecum of mice, which may favor the growth and gastrointestinal colonization of *Candida albicans* [73].

In conclusion, the role of SCFA (generated by the gastrointestinal or the upper airway microbiota) in preventing or facilitating pathogenic bacterial infections has been validated to be critical. The development of metabolite-mediated therapies, either through direct supplementation or induction of endogenous metabolites via diet-induced modifications of the microbiota, has been extensively researched [75,76]. Collectively, our study provides compelling evidence that propionic, butyric and caproic acids are important carbon sources promoting metabolism and growth of MAH, and it also creates exciting therapeutic opportunity to utilize SCFA for improving antibiotic effectiveness against persistent forms of MAH.

## Figures and Tables

**Figure 1 pathogens-09-00700-f001:**
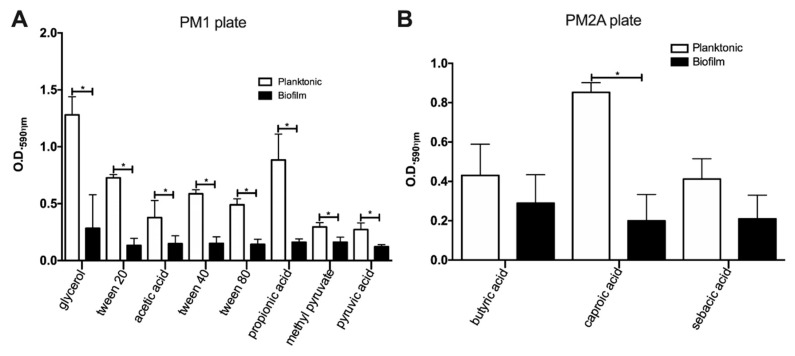
The utilization of carbon substrates by MAH104 in planktonic and biofilm states. The planktonic and biofilm cultures of MAH104 were tested for the ability to process various carbon-dependent metabolites from the Biolog Microplate™ plates. Both cultures were inoculated in PM1 (**A**) and PM2A (**B**) plates containing the inoculating media GN/GP-IF-0a supplemented with the appropriate additives and the Redox Dye Mix G and incubated at 37 °C for 7 days. Data represent the means ± standard deviations (SD) of results obtained from three independent experiments. Unpaired two tailed *t*-test was performed. * *p* < 0.05 was considered as statistically significant.

**Figure 2 pathogens-09-00700-f002:**
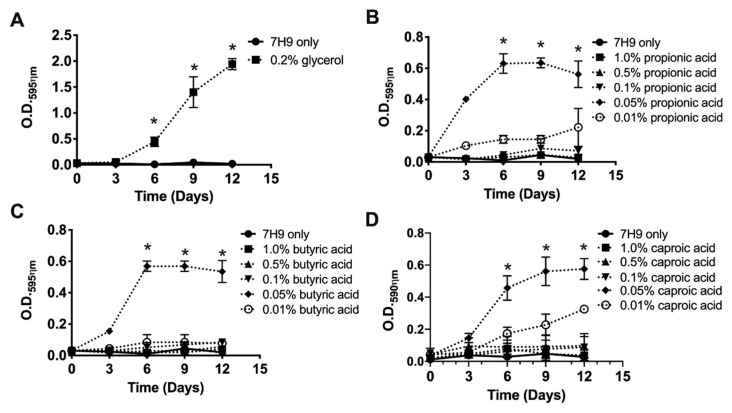
The effect of SCFA and glycerol on the growth of planktonic MAH104. Bacteria were cultured in non-supplemented 7H9 broth with glycerol (**A**), propionic acid (**B**), butyric acid (**C**) or caproic acid (**D**) under agitation for 12 days at 37 °C. Bacterial growth was recorded with optical density readings at O.D._595ηm_ at each 72 h. The non-supplemented 7H9 without substrates served as a negative (no growth) control (solid line). Data represent the means ± standard deviations (SD) of four independent experiments performed in triplicate. Two-way ANOVA followed by bonferroni post-test was Performed. * *p* < 0.05 was considered as statistically significant.

**Figure 3 pathogens-09-00700-f003:**
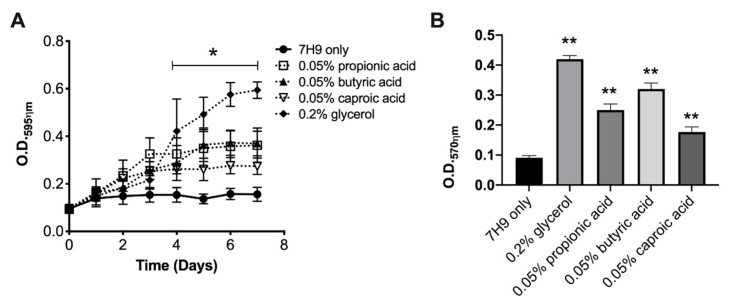
The effect of SCFA and glycerol on the biofilm formation of the sessile MAH104 of planktonic state. Planktonic bacteria at low density were seeded in 96-well polystyrene plates for 7 days. Next, supernatants were removed and reminder (surface-attached) bacteria were supplemented or not with 0.2% glycerol, 0.05% propionic acid, 0.05% butyric acid, and 0.05% caproic acid. (**A**) MAH104 growth were monitored though optical density (O.D._595ηm_) every 24 h over 7 days. Data represent the means ± standard deviations (SD) of four independent experiments performed with eight technical replicates. Two-way ANOVA analysis followed by bonferroni post-test was performed. * *p* < 0.05 was considered as statistically significant. (**B**) MAH104 biofilm mass were measured at day 7. Data represent the means ± standard deviations (SD) of four independent experiments performed with eight technical replicates. Unpaired two tailed *t*-test was made. ** *p* < 0.01 statistical significance between the substrate treated groups and the control.

**Figure 4 pathogens-09-00700-f004:**
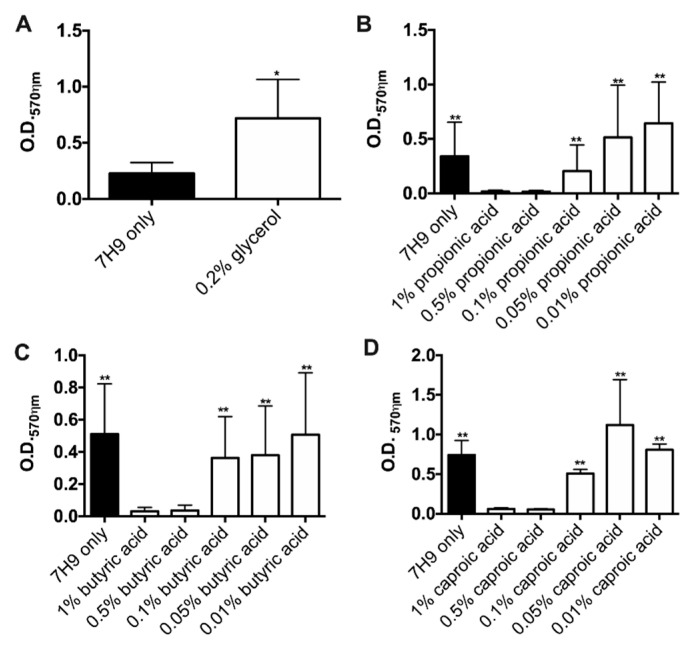
The effect of SCFA and glycerol on the static biofilm formation of MAH104. MAH104 static biofilms were formed by seeding 100 μL of suspension (1 × 10^8^ bacteria/mL) in 96 wells polystyrene plates and in non-supplemented 7H9 medium with or without glycerol (**A**), propionic acid (**B**), butyric acid (**C**) and caproic acid (**D**). The static biofilm formation was evaluated in range concentrations of targeted substrates using the crystal violet methodology at day 7. The data represent the means ± standard deviations (SD) of four independent experiments performed with eight technical replicates. MAH104 incubation with 0.5% and 1% of fatty acids (propionic acid, butyric acid and caproic acid) displayed a significant inhibition of bacterial biofilms in comparisons to other concentrations of the same substrate. Unpaired two tailed *t*-test was used. * *p* < 0.05 and ** *p* < 0.01.

**Figure 5 pathogens-09-00700-f005:**
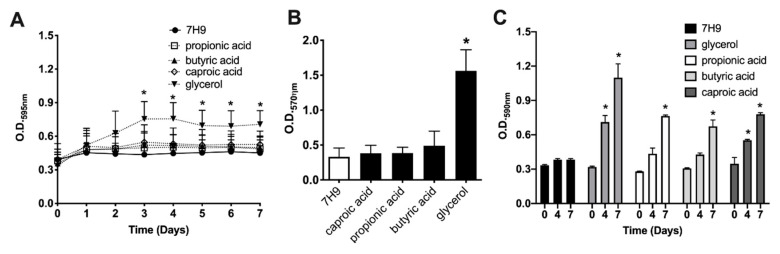
The effect of SCFA and glycerol on MAH104 growth within established biofilms. Seven day established biofilms of MAH104 were incubated with non-supplemented 7H9 with or without propionic acid, butyric acid, caproic acid and glycerol for 7 days at 37 °C. (**A**) Bacterial growth was recorded with optical density readings at O.D._595ηm_ every 24 h. Data represent the means ± standard deviations (SD) of three independent experiments performed in eight technical replicates. Two-way ANOVA analysis followed by bonferroni post-test was made. * *p* < 0.05 was considered statistically significant in comparison with negative control. (**B**) Biofilm formation was analyzed with crystal violet assay. Data represent the means ± standard deviations (SD) of three independent experiments performed in eight technical replicates. * *p* < 0.05 was considered as statistically significant between glycerol and control as well as the short-chain fatty acid experimental groups. For MAH104 metabolic screening (**C**), biofilms were established in the IF-0a GN/GP media for 7 days. Next, 100 μL of tested metabolites solubilized in the IF-0a GN/GP media were added to established biofilms at day 0 and incubated at 37 °C for additional 7 days. The Biolog Dye G Mix were added at day 0, 4 and 7 time-points and the spectrophotometric measurements were obtained after 24h at O.D._590ηm_. To reduce bacteria interference with readings, MAH104 tested plates were centrifuged for 20 min at 3500 rpm and supernatants were transferred into a new 96-well plate for O.D. The IF-0a GN/GP media with non-supplemented 7H9 served as a control. Data represent the means ± standard deviations (SD) of three independent experiments performed in eight technical replicates. Unpaired two tailed *t*-test was performed. * *p* < 0.05 was considered as statistically significant between metabolite experimental and control groups at corresponding time points.

**Table 1 pathogens-09-00700-t001:** List of metabolites used by MAH104 in planktonic and biofilm states.

Plate Well	Chemical Compound	Planktonic	Biofilm	Abiotic Reaction
**PM1 plate**				
A1	Negative control	-	-	-
B3	Glycerol	+	+	-
C5	Tween 20	+	+	-
C8	Acetic acid	+	+	-
D5	Tween 40	+	+	-
D6	α-Ketoglutaric acid	+	-	-
D7	α-Ketobutyric acid	+	-	-
E5	Tween 80	+	+	-
E7	α-Hydroxybutyric acid	+	-	-
F7	Propionic acid	+	+	-
G7	Acetoacetic acid	+	-	-
G9	Monomethyl succinate	+	-	-
G10	Methyl pyruvate	+	+	-
H8	Pyruvic acid	+	+	-
**PM2A plate**				
A1	Negative control	-	-	-
D12	Butyric acid	+	+	-
E2	Caproic acid	+	+	-
F8	Sebacic acid	+	+	-

Each test well was compared against negative control well of respective PM plate using unpaired *t*-test. Substrates with the signal intensity increase in 25% or more from negative control and a *p* < 0.05 were considered as a positive phenotype. The metabolites that were not metabolized by planktonic or biofilm cultures are not shown. The O.D._590ηm_ values are in the Appendix A.

**Table 2 pathogens-09-00700-t002:** In vitro antibiotic efficacy against blood and lung isolates of MAH in established biofilms supplemented with glycerol and short-chain acids.

Treatment	CFU/mL at 14 Days		
	MAH104	MAH3388	MAH3393
None	4.2 ± 0.5 × 10^8^	9.4 ± 0.2 × 10^7^	1.0 ± 0.8 × 10^8^
Clarithromycin	8.9 ± 0.8 × 10^7^	-	-
Amikacin	5.8 ± 0.5 × 10^7^	5.4 ± 0.6 × 10^6^	5.2 ± 0.6 × 10^7^
Glycerol	1.1 ± 0.3 × 10^9^	n/a	n/a
Glycerol + clarithromycin	1.7 ± 0.2 × 10^4^ *	-	-
Glycerol + amikacin	7.0 ± 0.2 × 10^5^ *	n/a	n/a
Propionic acid	5.5 ± 0.4 × 10^6^	9.6 ± 0.4 × 10^7^	1.7 ± 0.4 × 10^7^
Propionic acid + clarithromycin	2.9 ± 0.3 × 10^3^ *	-	-
Propionic acid + amikacin	3.6 ± 0.6 × 10^3^ *	3.1 ± 0.5 × 10^3^ *	1.6 ± 0.3 × 10^3^ *
Butyric acid	3.4 ± 0.6 × 10^6^	9.7 ± 0.2 × 10^7^	6.2 ± 0.4 × 10^7^
Butyric acid + clarithromycin	2.7 ± 0.5 × 10^3^ *	-	-
Butyric acid + amikacin	6.3 ± 0.7 × 10^3^ *	2.4 ± 0.5 × 10^3^ *	3.1 ± 0.5 × 10^3^ *
Caproic acid	6.4 ± 0.2 × 10^6^	8.4 ± 0.7 × 10^7^	4.7 ± 0.3 × 10^7^
Caproic acid + clarithromycin	4.9 ± 0.6 × 10^3^ *	-	-
Caproic acid + amikacin	7.3 ± 0.4 × 10^3^ *	4.1 ± 0.4 × 10^3^ *	1.6 ± 0.5 × 10^3^ *

* *p* < 0.05, between the antibiotic treatment groups in combination with metabolites and the with either the corresponding metabolite or corresponding antibiotic treatment group using unpaired *t*-test. Data represent the means ± standard deviations (SD) of three independent experiments performed in triplicates.

**Table 3 pathogens-09-00700-t003:** The antibiotic efficacy against intracellular MAH104 in THP-1 cells supplemented with glycerol and short-chain acids.

Treatment	CFU/mL Cell Lysate	
	2 h	4 Days
None	2.0 ± 0.4 × 10^5^	8.2 ± 0.3 × 10^5^
Clarithromycin		3.9 ± 0.3 × 10^4^ *
Amikacin		4.8 ± 0.5 × 10^4^ *
Glycerol	2.3 ± 0.2 × 10^5^	9.3 ± 0.3 × 10^5^
Glycerol + clarithromycin		8.3 ± 0.4 × 10^3^ *
Glycerol + amikacin		9.1 ± 0.5 × 10^3^ *
Propionic acid	3.1 ± 0.2 × 10^5^	8.0 ± 0.2 × 10^5^
Propionic acid + clarithromycin		1.1 ± 0.3 × 10^4^ *
Propionic acid + amikacin		1.7 ± 0.3 × 10^4^ *
Butyric acid	3.0 ± 0.3 × 10^5^	8.4 ± 0.5 × 10^5^
Butyric acid + clarithromycin		2.0 ± 0.4 × 10^4^ *
Butyric acid + amikacin		2.0 ± 0.3 × 10^4^ *
Caproic acid	2.4 ± 0.4 × 10^5^	5.3 ± 0.4 × 10^5^
Caproic acid + clarithromycin		3.6 ± 0.2 × 10^4^ *
Caproic acid + amikacin		1.6 ± 0.4 × 10^4^ *

* *p* < 0.05, between the antibiotic treatment groups in combination with metabolites and the corresponding metabolite or corresponding antibiotic treatment group alone, using unpaired *t*-test. Data represent the means ± standard deviations (SD) of three independent experiments performed in triplicates.

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
