# Peer review of "Short-Chain Fatty Acids Promote Mycobacterium avium subsp. hominissuis Growth in Nutrient-Limited Environments and Influence Susceptibility to Antibiotics"

_pathogens, 2020, doi:10.3390/pathogens9090700_

Round 1
Reviewer 1 Report
The manuscript describes the influence of short chain fatty acids on metabolic activity, growth, and antibiotic susceptability in M. avium. The results support the hypothesis that the presence of fatty acids, or species derived as fatty acid metabolites, can render the mycobacteria more suspectible to antibiotic treatment.
I could not completely understand the choice of "SCFA metabolites". Given that the authors describe C3, C4, and C6 acids, it might be useful to show the acids and the metabolites. The paper and the discussion go back and forth between the SCFA and their metabolites, making this somewhat confusing for the reader.
The discussion seems to largely avoid discussing the main theme of reference #49 (re influence of SCFA on M. avium): "has been only few explored." More accurately: "Recent work has suggested that SCFA may limit clearance of M. avium by alveolar macrophages. "
Within Table 1, column 2 is unclear. The heading is "Metabolite" yet the entries include Tween 20, Tween 40, Tween 80, and Sebacic acid (too large to be a metabolite)
Question: Abstract lines 23-24: Why is this "conversely", and not "Increased bacterial growth also observed under all conditions except?
Some minor corrections:
line 14: remodeling undergone by MAH in response to host environmental stresses or within biofilms formed in bronchial airways plays..
line 38: unclear. Do you mean that therapeutic options are, a a result, limited? Or do you intend there is a lack (in other words, no) therapeutic options.
Line 409: undergo cell surface modifications (delete "through")
Line 425: The current report is the first to describe processing of SCFA by MAH biofilms.
LIne 446: What is "capacity" for biofilm formation? If actual, then perhaps "extent" (or degree?) of biofilm formation.
Line 490: Glycerol promote(s)
Author Response
Reviewer 1.
I could not completely understand the choice of "SCFA metabolites". Given that the authors describe C3, C4, and C6 acids, it might be useful to show the acids and the metabolites. The paper and the discussion go back and forth between the SCFA and their metabolites, making this somewhat confusing for the reader.
A: The short-chain fatty acids (SCFA) are the metabolites. There is no difference between SCFA and “their metabolites”; both expressions describe the same term. To avoid this confision, we replacedwith SCFA (lines 21, 24, 84,495 and 561).
In the initial screening, we used the Biolog phenotype plates to identify chemical compounds that are metabolized by M. avium within biofilms (Table 1 and Figure 1). With this approach we found that M. avium was able to utilize eleven chemical compounds when in biofilms. In order to make the current study more feasible, we decided to further focus on compounds that have a potential role in in vivo during M. avium infection.
We know that M. avium colonizes the gastrointestinal tract and lungs [1], and microbiome of both organs produce several metabolites including short-chain fatty acids (SCFA) [2,3], which play a crucial role in human health and disease [2]. SCFA such as propionic acid, butyric acid and caproic acid can exert positive as well as negative effect on the growth of bacteria in in vivo environment [3-5]. Recent findings also suggest that SCFA might affect human susceptibility to mycobacterial infection (e.g., M. avium and Mycobacterium tuberculosis) [6-10]. These studies, however, explored the potential effect of SCFA on immune system against mycobacteria. The ability of M. avium to utilize SCFA within biofilms and their influence to activate M. avium metabolism and growth including impact on antibiotic susceptibility never been studied. Therefore, we decided to investigate the potential effect of propionic, butyric and caproic acids on the growth of M. avium when in biofilms. We included a brief explanation about the criteria used for choosing SCFA in lines 262-263 of the results section.
References
- Ratnatunga, C.N.; Lutzky, V.P.; Kupz, A.; Doolan, D.L.; Reid, D.W.; Field, M.; Bell, S.C.; Thomson, R.M.; Miles, J.J. The Rise of Non-Tuberculosis Mycobacterial Lung Disease. Frontiers in immunology 2020, 11, 303, doi:10.3389/fimmu.2020.00303.
- Kho, Z.Y.; Lal, S.K. The Human Gut Microbiome - A Potential Controller of Wellness and Disease. Frontiers in microbiology 2018, 9, 1835, doi:10.3389/fmicb.2018.01835.
- Ghorbani, P.; Santhakumar, P.; Hu, Q.; Djiadeu, P.; Wolever, T.M.S.; Palaniyar, N.; Grasemann, H. Short chain fatty acids affect cystic fibrosis airway inflammation and bacterial growth. European Respiratory Journal 2015, 46, 1033-1045, doi:10.1183/09031936.00143614
- Guinan, J.; Wang, S.; Hazbun, T.R.; Yadav, H.; Thangamani, S. Antibiotic-induced decreases in the levels of microbial-derived short-chain fatty acids correlate with increased gastrointestinal colonization of Candida albicans. Scientific reports 2019, 9, 8872, doi:10.1038/s41598-019-45467-7.
- Ciarlo, E.; Heinonen, T.; Herderschee, J.; Fenwick, C.; Mombelli, M.; Le Roy, D.; Roger, T. Impact of the microbial derived short chain fatty acid propionate on host susceptibility to bacterial and fungal infections in vivo. Scientific reports 2016, 6, 37944, doi:10.1038/srep37944.
- Bellerose, M.M.; Baek, S.H.; Huang, C.C.; Moss, C.E.; Koh, E.I.; Proulx, M.K.; Smith, C.M.; Baker, R.E.; Lee, J.S.; Eum, S., et al. Common Variants in the Glycerol Kinase Gene Reduce Tuberculosis Drug Efficacy. mBio 2019, 10, doi:10.1128/mBio.00663-19.
- Carpenito, J.W., B.; Sulaiman, I. ; Kurz, S.G.; Li, Y.; Perez, L.; Franca, B.; Olsen, E.; Gonzalez, A.; Yie, K.; Ma, S.; Naidoo, C.; Theron, G.; Weiden, M.; Basavaraj, A.; Tsay, J.J. ; Condos, R.; Kamelhar, D.L.; Addrizzo-Harris, D.J.; Segal, L.N. Microbial Short Chain Fatty Acids Impair Mycobacterium Avium (MAC) Clearance by Alveolar Macrophages. In Proceedings of American Thoracic Society 2019 International Conference, Dallas, TX., May 17 - May 22; p. A4242.
- Hicks, N.D.; Yang, J.; Zhang, X.; Zhao, B.; Grad, Y.H.; Liu, L.; Ou, X.; Chang, Z.; Xia, H.; Zhou, Y., et al. Clinically prevalent mutations in Mycobacterium tuberculosis alter propionate metabolism and mediate multidrug tolerance. Nature microbiology 2018, 3, 1032-1042, doi:10.1038/s41564-018-0218-3.
- Maji, A.; Misra, R.; Dhakan, D.B.; Gupta, V.; Mahato, N.K.; Saxena, R.; Mittal, P.; Thukral, N.; Sharma, E.; Singh, A., et al. Gut microbiome contributes to impairment of immunity in pulmonary tuberculosis patients by alteration of butyrate and propionate producers. Environmental microbiology 2018, 20, 402-419, doi:10.1111/1462-2920.14015.
- Safi, H.; Gopal, P.; Lingaraju, S.; Ma, S.; Levine, C.; Dartois, V.; Yee, M.; Li, L.; Blanc, L.; Ho Liang, H.P., et al. Phase variation in Mycobacterium tuberculosis glpK produces transiently heritable drug tolerance. Proceedings of the National Academy of Sciences of the United States of America 2019, 116, 19665-19674, doi:10.1073/pnas.1907631116.
The discussion seems to largely avoid discussing the main theme of reference #49 (re influence of SCFA on M. avium): "has been only few explored." More accurately: "Recent work has suggested that SCFA may limit clearance of M. avium by alveolar macrophages."
A: Yes, on this line we do not discuss the main theme of reference 49, however, we highlight this reference later in the text: “It has also been shown that metabolic products of anaerobic fermentation like butyrate and other SCFA impair macrophage ability to phagocytize and kill M. avium complex [44]” (lines 537 - 539). Thus, to avoid repetition, we would ask to leave the sentence as it is. We want to highlight that reference 49 is now reference 44.
Within Table 1, column 2 is unclear. The heading is "Metabolite" yet the entries include Tween 20, Tween 40, Tween 80, and Sebacic acid (too large to be a metabolite)
A: We agree with the reviewer. The word “metabolite” was replaced for “chemical compounds”. Thank you.
Question: Abstract lines 23-24: Why is this "conversely", and not "Increased bacterial growth also observed under all conditions except?
A: The reviewer is right. The word conversely was removed.
The original phrase was: “Selected SCFA metabolites enhanced MAH metabolic activity in planktonic and sessile states as well as in the static and established biofilms during nutrient-limited condition. Conversely, the increased bacterial growth was observed in all conditions except in established biofilms”.
The corrected sentence is: “Selected SCFA metabolites enhanced MAH metabolic activity in planktonic and sessile states as well as in the static and established biofilms during nutrient-limited condition. The increased bacterial growth was observed in all conditions except in established biofilms.” Line 23
Some minor corrections:
line 14: remodeling undergone by MAH in response to host environmental stresses or within biofilms formed in bronchial airways plays..
A: The correction has been made.
Previous sentence was: “The metabolic remodeling that MAH undergo under the host environmental stresses and within biofilms formed in bronchial airways plays an important role in development of the persistence phenotype contributing to the pathogen’s tolerance to antibiotic treatment.”
The modified sentence: “The metabolic remodeling of MAH in response to host environmental stresses or within biofilms formed in bronchial airways plays an important role in development of the persistence phenotype contributing to the pathogen’s tolerance to antibiotic treatment.” Line 14
line 38: unclear. Do you mean that therapeutic options are, as a result, limited? Or do you intend there is a lack (in other words, no) therapeutic options.
A: We made a change to avoid this confusion.
Original sentence: “The prolonged antibiotic regimens promote the development of acquired drug-resistance in MAH [7], and in some instances the lack of therapeutic options is quite common [6,8].”
The corrected sentence: “The therapeutic options for NTM patients are limited [6,7], and the prolonged antibiotic regimens required for MAH treatment can predispose the pathogen for the development of acquired drug-resistance [8]”. Please see lines 36-38. The reference 7 and 8 are currently references 8 and 7, respectively.
Line 409: undergo cell surface modifications (delete "through")
A: The word “through” was deleted.
Previous sentence was “Within the host, under extended exposure of antibiotics and environmental stresses, MAH undergo through cell surface modifications and metabolic remodeling, which results in the development of persistent phenotype [15].”
The corrected sentence now is “Within the host, under extended exposure of antibiotics and environmental stresses, MAH undergo cell surface modifications and metabolic remodeling, which results in the development of persistent phenotype [15].” This sentence is now found in lines 424-426.
Line 425: The current report is the first to describe processing of SCFA by MAH biofilms.
A: The sentence was rewritten as requested.
Previous phrase “Current work is the first study to demonstrate that MAH biofilms are able to process SCFA.”
Corrected sentence “Current report is the first to describe processing of SCFA by MAH biofilms.” This sentence is now in lines 441-442.
Line 446: What is "capacity" for biofilm formation? If actual, then perhaps "extent" (or degree?) of biofilm formation.
A: We decided to remove “the capacity of” from the text.
Previous sentence “Moreover, lower amount of the SCFA enhanced bacterial growth and the capacity of biofilm formation by sessile forms of MAH in nutrient-limited conditions.”
Now “Moreover, lower amount of the SCFA enhanced bacterial growth and biofilm formation by sessile forms of MAH in nutrient-limited conditions.” Now lines 461-462
Line 490: Glycerol promote(s)
A: The correction was made. Line 513.
Reviewer 2 Report
In this manuscript, the authors test the influence of short chain fatty acids on M. avium subsp hominissuis growth and susceptibility to antibiotics. They demonstrate that select short chain fatty acids enhance M avium subsp hominssuis growth and antibiotic susceptibility. The topic is of interest, the manuscript is well written and the experiments well designed with appropriate controls.
Comments:
-The methods section describes three MAH clinical strains used in the experiments, however it appears that the majority of the experiments only use strain "MAH/ MAH104", and only Table 2 includes data from all 3 strains. Is this correct? The term "MAH" seems to be used both generally to mean M. avium subsp hominissuis and also specifically for the strain from the blood of the AIDS patient. Suggest referring to the strain as "MAH104" throughout to avoid this confusion. Additionally, if only the one strain ("MAH/ MAH104") was used for the majority of of the experiments, this should be listed as a limitation of the study. If all three strains were used for all of the experiments, then variation between the strains in the results should be discussed.
-Data in Table 2 would perhaps be better presented in a figure that would allow for direct visualization of the impact of SCFAs in combination with the antibiotics on MAH growth.
-Discussion section lines 466-467 states that the SCFAs induced MAH killing in combination with clarithromycin across multiple isolates, whereas the data presented in Table 2 show that only one isolate (MAH104) was used for this finding (other 2 isolates did not have any growth in the setting of clarithromycin)
-The discussion section would benefit from the addition of a section on potential limitations of the study.
Author Response
Reviewer 2.
The methods section describes three MAH clinical strains used in the experiments, however it appears that the majority of the experiments only use strain "MAH/ MAH104", and only Table 2 includes data from all 3 strains. Is this correct? The term "MAH" seems to be used both generally to mean M. avium subsp hominissuis and also specifically for the strain from the blood of the AIDS patient. Suggest referring to the strain as "MAH104" throughout to avoid this confusion. Additionally, if only the one strain ("MAH/ MAH104") was used for the majority of of the experiments, this should be listed as a limitation of the study. If all three strains were used for all of the experiments, then variation between the strains in the results should be discussed.
A: Thank you for the suggestion. MAH104 strain was used in all assays, and MAH3388 and MAH3393 strains only in established biofilm assay presented in the table 2. We replaced the acronym MAH to MAH104 in the methodology and results sections. We also indicated in the legend of figures and tables as well as their titles which strain was used for each experiment. In lines 92 – 93 of methodology, we added sentence “The MAH104 strain was used in all assays, while MAH3388 and MAH3393 strains only when indicated.”
In the methodology (lines 198 – 201), it was also explained that “We also tested the ability of SCFA to enhance the efficacy of amikacin against biofilms formed by the MAH3388 and MAH3393 lung isolates. These strains are resistant to clarithromycin due to the presence of erm gene [42] and, thus, they were not subjected for clarithromycin treatment assays.”
Similar explanation was also included in the results (lines 388 – 391): “We also investigated whether SCFA could improve the amikacin efficacy against biofilms formed by the clarithromycin-resistant strains MAH3388 and MAH3393. These strains were not subjected for clarithromycin treatment assays.”
Finally, we made a minor modification in the lines 391-392 to make clearer that in table 2 the assays were made with all three MAH strains. Previous sentence: “Table 2 displays CFUs counts for three MAH clinical strains after clarithromycin and/or amikacin treatment in established biofilms…” The modified sentence: “Table 2 displays CFUs counts for three MAH clinical strains (MAH104, MAH3388, MAH3393) after clarithromycin and/or amikacin treatment in established biofilms…”
Data in Table 2 would perhaps be better presented in a figure that would allow for direct visualization of the impact of SCFAs in combination with the antibiotics on MAH growth.
A: We agree with the reviewer that the graph would have been a best presentation for the data, but because MAH3388 and MAH3393 strains could not be tested for clarithromycin treatment due to the resistance as well as glycerol assay was not performed for these two MAH strains, we believe that the graph will not properly display results. Here, we think that the table is the best way to show the data.
Discussion section lines 466-467 states that the SCFAs induced MAH killing in combination with clarithromycin across multiple isolates, whereas the data presented in Table 2 show that only one isolate (MAH104) was used for this finding (other 2 isolates did not have any growth in the setting of clarithromycin)
A: I apologize for the confusion. The sentence was modified.
Original sentence: “We found that all three fatty acids significantly induced killing of MAH clinical isolates by amikacin and/or clarithromycin in biofilms when compared with the viable bacterial number treated with only antibiotics or exposed only with metabolite supplements.”
Corrected sentence: “We found that all three fatty acids significantly induced killing of MAH104 by amikacin and clarithromycin in biofilms when compared with the viable bacterial number treated with only antibiotics or exposed only with metabolite supplements. In addition, we show that the SCFA compounds used in our study improved the efficacy of amikacin against the clarithromycin-resistant MAH3388 and MAH3393 clinical strains.” This sentence is now found in lines 482-487.
The discussion section would benefit from the addition of a section on potential limitations of the study.
A: We inserted a section in the discussion on potential limitations of this study as suggested by the reviewer. Lines 507-512.